# The Role of Ellagic Acid and *Annona muricata* in the Management of Human Papillomavirus (HPV)-Related Genital Lesions: A Systematic Review

**Tommaso Cai** [1,2,*], **Michele Rizzo** [3], **Giovanni Liguori** [3], **Michele Palumbo** [4], **Alessandro Palmieri** [5] **and Luca Gallelli** [6,7]

1   Department of Urology, Santa Chiara Hospital, 30121 Trento, Italy
2   Institute of Clinical Medicine, University of Oslo, 0315 Oslo, Norway
3   Department of Urology, University of Trieste, 24123 Trieste, Italy
4   Operative Unit of Urology, University of Catanzaro, 88100 Catanzaro, Italy
5   Department of Urology, University of Naples, Federico II, 80138 Naples, Italy
6   FAS@UMG Research Center Department of Health Science, University of Catanzaro, 88100 Catanzaro, Italy
7   Operative Unit of Clinical Pharmacology and Pharmacovigilance, Medifarmagen SRL,
    Mater Domini Hospital, 88100 Catanzaro, Italy
*   Correspondence: ktommy@libero.it

**Abstract:** *Background*: Several reports highlighted the role of nutraceutical compounds in the prevention and management of HPV-related genital lesions both in men and women, with interesting results. Here, we reviewed the effect of ellagic acid and *Annona muricata* for managing HPV-related genital lesions. *Methods*: Relevant databases were searched by using methods recommended by the Preferred Reporting Items for Systematic Reviews and Meta-Analysis guidelines. The primary endpoint was the clinical cure, defined as the clinical and/or laboratory and/or histopatologically demonstrated absence of HPV-related lesions at the end of the treatment. *Results*: We enclosed two retrospective studies, two prospective studies and one randomized controlled trial. In men, ellagic acid and *Annona muricata* complex improves seminal parameters and HPV-DNA clearance. In women, it has a chemopreventive action in cervical cancer and increases the HPV viral clearance. No clinically significant adverse effects have been reported. *Conclusions*: In conclusion, the combination of ellagic acid and *Annona muricata* shows interesting and promising results in terms of HPV viral clearance and HPV related genital lesions. However, more data are necessary to confirm these results.

**Keywords:** *Annona muricata*; ellagic acid; human papilloma virus; HPV infections; efficacy; safety





## 1. Introduction

Human papilloma virus (HPV) has been recognized as one of the most common sexually transmitted virus infections in men and women, including both oncogenic and non-oncogenic viruses [1,2]. HPV infections are associated with benign diseases, such as genital warts and malignancies such as cervical, oropharynx and penile cancers [3,4]. The persistence of high-risk HPV genotypes could lead to the progression of cervical intraepithelial neoplasia [5], and the aim of the treatment should be the reduction in the time for the HPV-DNA clearance rate and the persistence of infection. Up to date, the management of HPV is suggested in the presence of genital, anorectal or dermatological clinical manifestations, such as genital or anorectal warts. No systemic drugs are available for the management of HPV infections. On the one hand, the first vaccine against HPV has been introduced in clinical practice in 2006 and now represents the most important preventive drug for HPV infections [6]. On the other hand, several experiences have been reported about the role of some polyphenols as well as *Annona muricata* and ellagic acid on cell-cycle arrest in G1 phase, DNA reparation and apoptosis in cancer cells [7–9].

Ellagic acid is a polyphenol compound found in many fruits and nuts [10,11], able to induce several biological effects, e.g., anti-inflammatory, anti-allergy, anti-bacterial [12,13] and anti-tumor properties [14,15]. In oral epithelial cell cultures [16], it was reported that ellagic acid can modulate the expression of innate immune mediators. Some authors started to use ellagic acid for managing male infertility. In this sense, Bucak et al. demonstrated that ellagic acid reduces the oxidative stress parameters in rat semen, highlighting its probable protective role in spermatozoa [11]. This activity is related to the antioxidant activity of ellagic acid, and it is able to deactivate peroxyl radicals, hydroxyl radicals, nitrogen dioxide and peroxynitrite [17]. Recently Kullappan et al., through molecular docking and molecular dynamic studies, documented that ellagic acid has the most favorable binding energy activity against Zika virus NS3 helicase, suggesting that this compound could be able to reduce the activity of the Zika virus [18]. Similarly, the effect of ellagic acid in solution and as a gel formulation on HIV enzymes in vitro, as well as HIV-1 replication and cytokines production in HIV-1-infected cells [19], was evaluated. In this study, the authors demonstrated that ellagic acid both in solution and in gel possessed potent anti-HIV-1 activity through the inhibition of the integrase enzyme HIV-1 replication, without the development of cytotoxicity. However, the authors also reported that ellagic acid administration did not reduce the secretion of IL-6 and IL-8 induced by HIV-infection.

*Annona muricata* (Annonaceae family) is known as "Soursop" or "Graviola" is a terrestrial deciduous tree used in several traditional medicines for its potent biomedical properties able to treat various diseases, such as stomach pain, bronchitis and gastric cancer [20]. The Annonaceae family, besides producing polyphenols, flavonoids and alkaloids, produces large quantities of acetogenins, a secondary metabolite derived from long-chain fatty acids [21] that have been found to have anticancer, anti-inflammatory, antibacterial and antiviral properties [22]. The phenolic compounds in *Annona muricata*, such as quercetin and gallic acid, are reported to be the compounds most responsible for the antioxidant capacity of the plant [22]. *Annona muricata* extracts quercetin and gallic acid, the compounds reported to be most responsible for antioxidant activity [22]; the flavonoids rutin (quercetin 3-O-rutinoside; 2) and naringenin (5,7-dihydroxy-2-(4-hydroxyphenyl) chroman-4-one; 3) are responsible for the antiviral activity because they can decrease viral replication [23]. Rutin is an effective inhibitor of dihydrofolate reductase and shows antiviral, anticancer, anti-inflammatory and heart disease protective activities [24]. Finally, acetogenins documented significant antiviral activities against herpes simplex virus–I (HSV-I) [25], dengue virus type 2 [26] and human immunodeficiency virus–I (HIV-I) [27]. In vivo studies have shown that rutin is an excellent candidate to eliminate the viral replication of SARS-CoV-2. Elmi et al. [28] suggest that rutin could inhibit angiotensin-converting enzyme 2 (ACE 2), the receptor that facilitates the entry of SARS-CoV-2 into lung cells, suppressing the infection.

The aim of this study is to review all available data on the role of ellagic acid and *Annona muricata* complex in the management of HPV-related genital infections.

### Research Question

We put forth this research query:

Is ellagic acid and *Annona muricata* complex able to obtain significant pre-clinical data and a clinical cure in men and women affected by HPV-related genital infections?

In order to respond to these research questions, we performed a systematic review of all available studies performed with the aim to evaluate the efficacy of ellagic acid and *Annona muricata* complex in the management of HPV infections both in males and females.

## 2. Materials and Methods

### Research Strategy and Literature Search

From June to November 2022, two independent reviewers (T.C., M.R.) performed studies in PubMed database, Cochrane CENTRAL and Scopus. (Figure 1) All disagreements between the two reviewers were resolved by a supervisor (A.P.). All references cited in

relevant articles were also reviewed and analyzed. The search strategy used was '(*Annona muricata*) AND/OR (ellagic acid) AND (human papillomavirus OR HPV)'. As filters, we used: clinical trial, humans, English language and adult. Moreover, we included all clinical trials evaluating the clinical efficacy of ellagic acid and *Annona muricata* complex tested for managing HPV infections both in males and females in this review. Titles and abstracts were used to screen for initial study inclusion. Full-text review was used where abstracts were insufficient to determine if the study met inclusion or exclusion criteria. Two authors (T.C., M.R.) independently performed all data abstraction, including evaluation of the study characteristics, risk of bias and outcome measures, with independent verification performed by the senior author. The study has been performed in line with the Preferred Reporting Items for Systematic Reviews and Meta-Analyses (PRISMA), the recommendations of the European Association of Urology Guidelines office for conducting systematic reviews and meta-analysis and previous studies [29,30]. All selected pre-clinical and clinical trials were used for the systematic review.

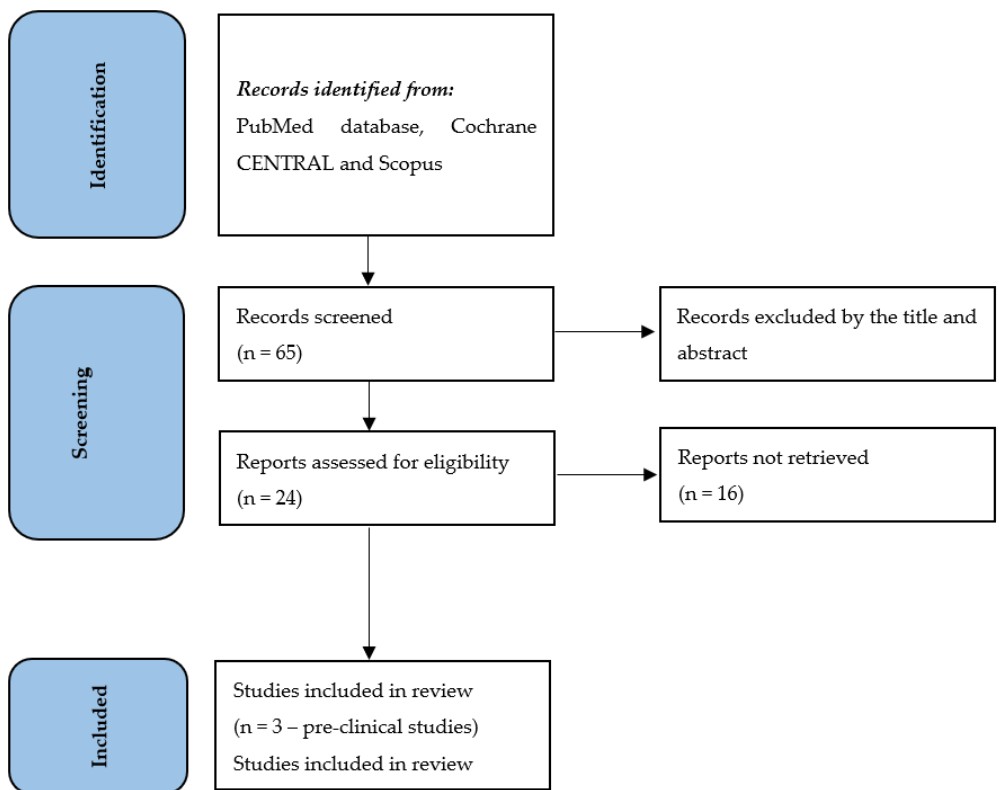

**Figure 1.** The flowchart of the literature search in line with the PRISMA statement.

### 3. Research Evidence

*3.1. Pre-Clinical Data*

We evaluated the antiviral effects of *Annona muricata* and ellagic acid. No pre-clinical studies have been performed on the combination of *Annona muricata* and ellagic acid. However, we found several pre-clinical studies evaluating the antiviral effect of *Annona muricata* or ellagic acid. The first study describing the protective effect of ellagic acid against the genotoxic effects of mutagenic agents was published in 1990 [31]. Moktar et al. [32], in an experimental study in HPV16-transformed human ectocervical cell lines, documented that cigarette smokers infected with HPV may be at a higher risk for cervical cancer. In this model, the treatment with ellagic acid (5 or 15 μM, 0.2% DMSO) for 20 h, was able to reduce DNA strand breaks induced by cigarette smoke in the cervical cells, supporting the data related to the antioxidant activity of ellagic acid. Park et al. [33], in the human epithelial adenocarcinoma cervix cell line infected with human rhinovirus, documented that ellagic acid significantly suppressed human rhinovirus infection only when added just

after the virus inoculation (0 h), but not 1 h before or after. These data suggest that ellagic acid interacts with the human cells in the early stage of human rhinoviruses infections to protect the cells from virus destruction. In this study, the authors documented that ellagic acid inhibits virus replication by targeting cellular molecules, rather than virus molecules [33]. On the other hand, *Annona muricata* extracts seem to have an interesting role in viral infections due to their capability to decrease viral replication [34]. All included pre-clinical studies in this review have been shown in the Table 1.

**Table 1.** The table shows the summary of all included pre-clinical studies in this review.

| Author | Year | Type of Study | Aim | Findings Description |
|---|---|---|---|---|
| Moktar A. [32] | 2009 | In vitro study | Evaluate the Ellagic acid effect as DNA breaks protector. | Ellagic acid is able to inhibit the DNA breaks in cells infected by HPV infection and treated with cigarette smoke condensate. |
| Park SW. [33] | 2014 | In vitro study | Evaluate the Ellagic acid effect in protecting the cells from virus infection. | Ellagic acid interacts with the human cells in the early stage of human rhinoviruses infections to protect the cells from the virus destruction. |
| Balderrama-Carmona AP. [34] | 2021 | In vitro study | To study the antiviral and antioxidant properties of *Annona muricata*. | Annona muricata seems to have a positive effect on reducing oxidative stress in human erythrocytes and viral infections. |

### 3.2. Clinical Data

We reviewed the data on studies investigating the clinical efficacy of the combination of *Annona muricata* and ellagic acid in patients affected by HPV infections.

#### 3.2.1. Effects in Men with HPV Infections

In male patients, ellagic acid and *Annona muricata* complex improve HPV-DNA clearance and reduce the time to the virus clearance. In a pilot study, Cai et al. [35] evaluated, for the first time, the effects of a 3-month treatment with ellagic acid and *Annona muricata* in 43 men with oligospermia and HPV-related infections. The authors documented an improvement of both the number of spermatozoa (10.6 vs. 15.8) and their mobility (27.5% vs. 36.1%). The authors reported a strong correlation between the HPV-DNA clearance rate and the number of spermatozoa (r = 0.78; *p* < 0.001). They concluded that ellagic acid and *Annona muricata* complex improved HPV-DNA clearance and seminal parameters, highlighting the role of this compound in the redirecting the immune responses in viral infections [35]. These results have been confirmed by La Vignera et al. [36] in males with infertility treated with ellagic acid (100 mg/day) and *Annona muricata* (100 mg/day) for 3 months, in comparison with patients under active surveillance (protected sexual intercourse). The authors found that ellagic acid and *Annona muricata* complex improve semen quality and HPV-DNA clearance rate in patients with HR-HPV infection [36].

#### 3.2.2. Effects in Pre-Neoplastic and Neoplastic Cervical Lesions

Several authors documented that ellagic acid and *Annona muricata* shows a chemopreventive action in women with cervical cancer, increasing the viral clearance of HPV [37–39]. In a prospective study, Morosetti et al. [37] documented the protective effects of ellagic acid and *Annona muricata* complex on viral clearance and/or cytological lesions in women with atypical cells of undetermined significance (ASCUS), low-grade squamous intraepithelial lesions (LSIL) or high-grade squamous intraepithelial lesions (HSIL). In a retrospective parallel cohort study, Moscato et al. [38] demonstrated that ellagic acid and *Annona muricata* complex induce apoptosis, cell cycle arrest and repair mechanisms in cervical cells. Interesting results have been reported, also, by Le Donne et al. [39] in a pilot study that investigated the anti-viral activity of ellagic acid and *Annona muricata* complex in women affected by low squamous intraepithelial lesions (L-SIL) related to high-risk human papilloma virus. These

trials documented that the complex ellagic acid and *Annona muricata* have an interesting role in terms of HPV viral clearance. All included clinical studies in this review have been shown in the Table 2.

**Table 2.** The table shows the summary of all included clinical studies in this review.

| Author | Year | Type of Study | Aim | Findings Description |
|---|---|---|---|---|
| Cai T. [35] | 2022 | Phase I-II study | To evaluate the effects of a 3-month treatment with ellagic acid and *Annona muricata* in men with oligospermia and HPV-related infections. | Ellagic acid and *Annona muricata* complex improved HPV-DNA clearance and seminal parameters. |
| La Vignera S. [36] | 2022 | Retrospective case–control study | To evaluate the effects of ellagic acid and *Annona muricata* in men with inefrtility and HPV infections. | Ellagic acid and *Annona muricata* complex improve semen quality and HPV-DNA clearance rate in patients with high-risk HPV infection |
| Morosetti G. [37] | 2017 | Phase III study | To evaluate the effect of ellagic acid and *Annona Muricata* on viral clearance and/or cytological lesions in a group of women with HPV cervical infection. | Women treated with ellagic acid and *Annona muricata* complex were less likely to be diagnosed with an abnormal Pap smear at 6 and 12 months when compared with the control group. |
| Moscato GMF. [38] | 2022 | Retrospective study | To compares the evolution of persistent cervical HPV infection in women treated with nonavalent vaccine or ellagic acid and *Annona muricata*. | The use of ellagic acid and *Annona muricata* seems an interesting clinical strategy in terms of increasing chance of HPV viral clearance. |
| Le Donne M. [39] | 2017 | Pilot study | To investigate the anti-viral activity of ellagic acid and *Annona muricata* complex in women affected by low squamous intraepithelial lesion (L-SIL) related to HPV infections. | Ellagic acid and *Annona muricata* complex is able to improve the HPV viral clearance. |

## 4. Conclusions, Limitations and Future Perspectives

Several studies investigating the role of diet and nutritional status in HPV genital infections suggested that natural compounds provide a protective effect against cervical dysplasia and HPV persistence [40,41]. Other authors demonstrated the antioxidant and chemopreventive activity of ellagic acid [42] and *Annona muricata* complex, focusing on its capability to induce cell cycle arrest in the G1 phase, DNA repair and apoptosis [7,8]. In this review, we demonstrated that medical devices containing ellagic acid and *Annona Muricata* show interesting and promising results in terms of HPV viral clearance and HPV-related genital lesions both in male and female patients. However, some limitations should be considered when analyzing these results. The limitations are: the small number of studies and the low number of enrolled patients. These limitations notwithstanding, our systematic review suggests the need of evaluating the effect of nutraceuticals and phytotherapic compounds as drugs. In particular, the use of a medical device containing ellagic acid and *Annona Muricata* could represent an interesting therapy to use in males and females with HPV genital infections for reducing the time to HPV viral clearance, in association with local therapy or the HPV vaccine. Long-term follow-up and further multicentric series would be necessary to definitively establish the long-term benefits of ellagic acid and *Annona muricata* complex. Future studies are needed to confirm these results.

**Author Contributions:** Conceptualization, T.C.; methodology, T.C. and M.R.; software, T.C.; formal analysis, T.C., M.R. and G.L.; data curation, M.R. and M.P.; writing—original draft preparation, T.C.; writing—review and editing, L.G.; supervision, A.P. All authors have read and agreed to the published version of the manuscript.

**Funding:** This research received no external funding.

**Institutional Review Board Statement:** Not applicable.

**Informed Consent Statement:** Not applicable.

**Data Availability Statement:** Not applicable.

**Conflicts of Interest:** The authors declare no conflict of interest.

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
