# Peer review of "The Role of Ellagic Acid and Annona muricata in the Management of Human Papillomavirus (HPV)-Related Genital Lesions: A Systematic Review"

_2673-4397, doi:10.3390/uro3010008_

Round 1
Reviewer 1 Report
Cai et al contribute a review on the relationship between ellagic acid and annona muricata and lesions caused by human papillomavirus. The review and objectives are clearly defined and formulated with a brief introduction to the HPV virus followed by an introduction of the two substances. The area described is, by all accounts, very small - the review covers most of the evidence in this area and raises the question of efficacy and use in the clinical setting.
Concept comments:
Even though this is only a little known topic and there is still a lot of knowledge to obtain in the future, there are some aspects of the review that need to be improved.
Major comments:
- A flowchart of the literature search would be a good visual support to with conprehend iterature search.
- The authors comment on inclusion and exclulsion criteria in line 106 - these do not seem to be mentioned. Insertion would be requiered.
- Maybe one could include PMID 2165562 which already described the influence of EA on papillomavirus-DNA in 1990.
- Goodwin et al (PMID 19297472) described the effect of chemicals that inhibti infection by the polyomavirus. Ellagic acid inhibited cell binding in their experiments which also should be implemented at least in the introduction section. Addtionally PMID 34726804 showes that e.g. ellagic acid has cytotoxic effects.
- The authors have formulated two research questions in the first section of their work. The conclusion yet does not mention or disuss potential hypothesis to the clinical practice or managment of applicable therapies in future? Clinical interpretation of the data up to now? Are there therapies known including the two mentioned substances? The conclusion needs to mark out the futher needed research strategies to move towards clinical implication.
Minor comments:
- line 43-44 - grammar - singular vaccine? there are more than one availabe
- line 44 - style - 'on the other hand' need a 'on the one hand' sometime bevor .. or rephrase
- line 50 - style - pronoun is missing
- line 66 - no fullstop behind 'Graviola'
- line 90 - style - the both subtances themselves will not be able to obtain clinical data or a cure - streamline English style
- line 184 'long-term'
- whole document - streamline English style and improve punctuation, especially commas
Author Response
Cai et al contribute a review on the relationship between ellagic acid and annona muricata and lesions caused by human papillomavirus. The review and objectives are clearly defined and formulated with a brief introduction to the HPV virus followed by an introduction of the two substances. The area described is, by all accounts, very small - the review covers most of the evidence in this area and raises the question of efficacy and use in the clinical setting.
Concept comments:
Even though this is only a little known topic and there is still a lot of knowledge to obtain in the future, there are some aspects of the review that need to be improved.
Major comments:
- A flowchart of the literature search would be a good visual support to with conprehend iterature search.
- Many thanks for your suggestion. A flowchart of the literature search has been added to the text.
- The authors comment on inclusion and exclulsion criteria in line 106 - these do not seem to be mentioned. Insertion would be requiered.
- Many thanks for your comment. In order to clarify the inclusion and exclusion criteria used the sentence following sentence has been added to the text: “Moreover, we included in this review all clinical trial evaluating the clinical efficacy of ellagic acid and Annona muricata complex tested for managing HPV infections both in males and females”.
- Maybe one could include PMID 2165562 which already described the influence of EA on papillomavirus-DNA in 1990.
- In line with your suggestion, the following sentence and reference has been added to the text: “The first study describing the protective effect of ellagic acid against the genotoxic effects of mutagenic agents, has been published in 1990 [31] - 31. Stich, H.F.; Tsang, S.S.; Palcic, B. The effect of retinoids, carotenoids and phenolics on chromosomal instability of bovine papillomavirus DNA-carrying cells. Mutat Res 1990 241(4):387-93.”.
- Goodwin et al (PMID 19297472) described the effect of chemicals that inhibti infection by the polyomavirus. Ellagic acid inhibited cell binding in their experiments which also should be implemented at least in the introduction section. Addtionally PMID 34726804 showes that e.g. ellagic acid has cytotoxic effects.
- in line with your suggestion the following sentence has been added to the discussion: “42. Pani, S.; Mohapatra, S.; Sahoo, A.; Baral, B.; Debata, P.R. Shifting of cell cycle arrest from the S-phase to G2/M phase and downregulation of EGFR expression by phytochemical combinations in HeLa cervical cancer cells. J Biochem Mol Toxicol 2022 36(1):e22947”.
- The authors have formulated two research questions in the first section of their work. The conclusion yet does not mention or disuss potential hypothesis to the clinical practice or anagement of applicable therapies in future? Clinical interpretation of the data up to now? Are there therapies known including the two mentioned substances? The conclusion needs to mark out the futher needed research strategies to move towards clinical implication.
- In line with your suggestions, the following sentences has been added to the discussion: “In particular, the use of a medical device containing ellagic acid and Annona Muricata could represent an interesting therapy to use in male and female with HPV genital infections in order to reduce the HPV viral clearance in association with local therapy or HPV vaccine”.
Minor comments:
- line 43-44 - grammar - singular vaccine? there are more than one availabe
- Many thanks for this suggestion.
- line 44 - style - 'on the other hand' need a 'on the one hand' sometime bevor .. or rephrase
- Many thanks for this suggestion.
- line 50 - style - pronoun is missing
- Many thanks for this suggestion.
- line 66 - no fullstop behind 'Graviola'
- Many thanks for this suggestion.
- line 90 - style - the both subtances themselves will not be able to obtain clinical data or a cure - streamline English style
- Many thanks for this suggestion.
- line 184 'long-term'
- Many thanks for this suggestion.
- whole document - streamline English style and improve punctuation, especially commas.
- Many thanks for this suggestion.
Reviewer 2 Report
Authors should be congratulated for their work. The topic is interesting. Tables are clear. The manuscript is well-written and easily readable. The different types of treatment for lesions of a very diffused infection like HPV is a huge topic to debate.
The manuscript is suitable for publication
Author Response
Authors should be congratulated for their work. The topic is interesting. Tables are clear. The manuscript is well-written and easily readable. The different types of treatment for lesions of a very diffused infection like HPV is a huge topic to debate.
The manuscript is suitable for publication
- Many thanks for your comments to our work.
Round 2
Reviewer 1 Report
All revision points have been dealt with in terms of content - yet English proofreading should be carried out to finalise the legibility of the work.
Author Response
All revision points have been dealt with in terms of content - yet English proofreading should be carried out to finalise the legibility of the work.
A. Many thanks for your comment. The manuscript has been revised by a native English speaker.